# LASER: Linear Compression in Wireless Distributed Optimization

## Abstract

Data-parallel SGD is the de facto algorithm for distributed optimization, especially for large scale machine learning. Despite its merits, communication bottleneck is one of its persistent issues. Most compression schemes to alleviate this either assume noiseless communication links, or fail to achieve good performance on practical tasks. In this paper, we close this gap and introduce LASER: **L**ine**A**r Compre**S**sion in Wir**E**less Dist**R**ibuted Optimization. LASER capitalizes on the inherent low-rank structure of gradients and transmits them efficiently over the noisy channels. Whilst enjoying theoretical guarantees similar to those of the classical SGD, LASER shows consistent gains over baselines on a variety of practical benchmarks. In particular, it outperforms the state-of-the-art compression schemes on challenging computer vision and GPT language modeling tasks. On the latter, we obtain 50-64% improvement in perplexity over our baselines for noisy channels.

## 1 Introduction

Distributed optimization is one of the most widely used frameworks for training large scale deep learning models (Bottou et al., 2018; Dean et al., 2012; Tang et al., 2020). In particular, data-parallel SGD is the workhorse algorithm for this task. Underpinning this approach is the *communication* of large gradient vectors between the workers and the central server which performs their *aggregation*. While these methods harness the inherent parallelism to reduce the overall training time, their communication cost is a major bottleneck that limits scalability to large models. Design of communication-efficient distributed algorithms is thus a must for reaping the full benefits of distributed optimization (Xu et al., 2020).

Existing approaches to reduce the communication cost can be broadly classified into two themes: (i) compressing the gradients before transmission; or (ii) utilizing the communication link for native 'over-the-air' aggregation (averaging) across workers. Along (i), a number of gradient compression schemes have been designed such as quantization (Bernstein et al., 2018; Vargaftik et al., 2022), sparsification (Aji & Heafield, 2017; Isik et al., 2022), hybrid methods (Jiang et al., 2018; Basu et al., 2019), and low-rank compression (Wang et al., 2018; Vogels et al., 2019). These methods show gains over the full-precision SGD in various settings (Xu et al. (2020) is a detailed survey). Notwithstanding the merits, their key shortcoming is that they assume a *noiseless* communication link between the clients and the server. In settings such as federated learning with differential privacy or wireless communication, these links are noisy. Making them noiseless requires error-correcting codes which exacerbates the latency, as the server needs to wait till it receives the gradient from each worker before aggregating (Guo et al., 2020).

Under theme (ii), communication cost is reduced by harnessing the physical layer aspects of (noisy) communication. In particular, the superposition nature of wireless channels is exploited to perform over-the-air averaging of gradients across workers, which reduces the latency, see e.g. Shi et al. (2020) and the references therein. Notable works include A-DSGD Amiri & Gündüz (2020b), analog-gradient-aggregation Guo et al. (2020); Zhu et al. (2019), channel aware quantization Chang & Tandon (2020), etc. However, to the best of our knowledge, the majority of these approaches are restricted to synthetic datasets and shallow neural networks (often single layer) and do not scale well to the practical neural network models (which we verify in Sec. 4). This leads to a natural question:

*Can we design efficient and practical gradient compression schemes for noisy communication channels?*

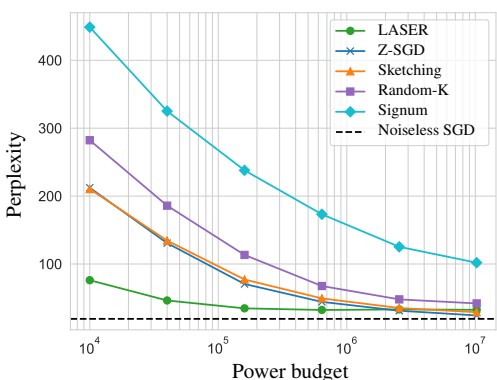

**Figure 1:** Final test perplexity after 20k iterations *(lower is better)* vs. power budget for GPT-2 language modeling on WIKITEXT-103. LASER consistently requires orders-of-magnitude less power than other methods for the same perplexity.

**Table 1:** Power required *(lower is better)* to reach the target perplexity on WIKITEXT-103. Z-SGD sends the uncompressed gradients directly, while LASER sends a rank-4 approximation. LASER requires $16\times$ less power than Z-SGD to achieve the target perplexity over a wide interval. In the very-high-power regime with perplexity close to that of the noiseless SGD, we see no power gains.

| Target | Power required | | Reduction |
|---|---|---|---|
| | Z-SGD | LASER | |
| 80 | 160 K | 10 K | $16\times$ |
| 50 | 640 K | 40 K | $16\times$ |
| 40 | 2560 K | 160 K | $16\times$ |
| 35 | 2560 K | 160 K | $16\times$ |

In this work, we precisely address this and propose LASER, a principled gradient compression scheme for distributed training over wireless noisy channels. Specifically, we make the following contributions:

- Capitalizing on the inherent low-rank structure of the gradients, LASER efficiently computes these low-rank factors and transmits them reliably over the noisy channel while allowing the gradients to be averaged in transit (Sec. 3).

- We show that LASER enjoys similar convergence rate as that of the classical SGD for both quasi-convex and non-convex functions, except for a small additive constant depending on the channel degradation (Theorem 1).

- We empirically demonstrate the superiority of LASER over the baselines on the challenging tasks of (i) language modeling with GPT-2 $\rightarrow$ WIKITEXT-103 and (ii) image classification with RESNET18 $\rightarrow$ (CIFAR10, CIFAR100) and 1-LAYER NN $\rightarrow$ MNIST. With high gradient compression ($165\times$), LASER achieves 50-64% perplexity improvement in the low and moderate power regimes on WIKITEXT-103. To the best of our knowledge, LASER is the first to exhibit such gains for GPT language modeling (Sec. 4).

**Notation.** Euclidean vectors and matrices are denoted by bold letters $\boldsymbol{x}, \boldsymbol{y}, \boldsymbol{M}$, etc. $\|\cdot\|$ denotes the Frobenius norm for matrices and the $\ell_2$-norm for Euclidean vectors. $\mathcal{O}(\cdot)$ is an upper bound subsuming universal constants whereas $\widetilde{\mathcal{O}}(\cdot)$ hides any logarithmic problem-variable dependencies.

## 2 BACKGROUND

**Distributed optimization**. Consider the (synchronous) data-parallel distributed setting where we minimize an objective $f : \mathbb{R}^d \rightarrow \mathbb{R}$ defined as the empirical loss on a global dataset $\mathcal{D} = \{(\boldsymbol{x}_j, y_j)\}_{j=1}^N$:

$$\min_{\boldsymbol{\theta} \in \mathbb{R}^d} f(\boldsymbol{\theta}), \quad f(\boldsymbol{\theta}) \triangleq \frac{1}{N} \sum_{j=1}^N \ell(\boldsymbol{x}_j, y_j; \boldsymbol{\theta}),$$

where $\ell(\cdot)$ evaluates the loss for each data sample $(\boldsymbol{x}_j, y_j)$ on model $\boldsymbol{\theta}$. In this setup, there are $k$ (data-homogeneous) training clients, where the $i^{\text{th}}$ client has access to a stochastic gradient oracle $\boldsymbol{g}_i$, e.g. mini-batch gradient on a set of samples randomly chosen from $\mathcal{D}$, such that $\mathbb{E}[\boldsymbol{g}_i | \boldsymbol{\theta}] = \nabla f(\boldsymbol{\theta})$ for all $\boldsymbol{\theta} \in \mathbb{R}^d$. In distributed SGD (Robbins & Monro, 1951; Bottou et al., 2018), the server aggregates all $\boldsymbol{g}_i$s and performs the following updates:

$$\boldsymbol{\theta}_{t+1} = \boldsymbol{\theta}_t - \gamma_t \cdot \frac{1}{k} \sum_{i=1}^k \boldsymbol{g}_i^{(t)}, \quad \mathbb{E}[\boldsymbol{g}_i^{(t)} | \boldsymbol{\theta}_t] = \nabla f(\boldsymbol{\theta}_t), \quad t \geq 0, \tag{SGD}$$

where $\{\gamma_t\}_{t\geq 0}$ is a stepsize schedule. Implicit here is the assumption that the communication link between the clients and the server is noiseless, which we expound upon next.

**Communication model.** For the communication uplink from the clients to the server, we consider the standard wireless channel for over-the-air distributed learning (Amiri & Gündüz, 2020a; Guo et al., 2020; Zhu et al., 2019; Chang & Tandon, 2020; Wei & Shen, 2022a): the *additive slow-fading channel*, e.g., the classical multiple-access-channel (Nazer & Gastpar, 2007). The defining property of this family is the superposition of incoming wireless signals (enabling over-the-air computation) possibly corrupted together with an independent channel noise (Shi et al., 2020). Specifically, we denote the channel as a (random) mapping $\mathcal{Z}_P(\cdot)$ that transforms the set of (time-varying) messages transmitted by the clients $\{\boldsymbol{x}_i\}_{i\in[k]} \subset \mathbb{R}^d$ to its noisy version $\boldsymbol{y} \in \mathbb{R}^d$ received by the server:

$$\boldsymbol{y} = \mathcal{Z}_P(\{\boldsymbol{x}_i\}) \triangleq \sum_{i=1}^{k} \boldsymbol{x}_i + \boldsymbol{Z}, \quad \|\boldsymbol{x}_i\|^2 \leq P_t, \; \frac{1}{T}\sum_{t=0}^{T-1} P_t \leq P, \tag{1}$$

where the noise $\boldsymbol{Z} \in \mathbb{R}^d$ is independent of the channel inputs and has zero mean and unit variance per dimension, i.e. $\mathbb{E}\|\boldsymbol{Z}\|^2 = d$. The power constraint on each client $\|\boldsymbol{x}_i\|^2 \leq P_t$ at time $t$ serves as a communication cost (and budget), while the power policy $\{P_t\}$ allots the total budget $P$ over $T$ epochs as per the average power constraint (Wei & Shen, 2022b; Amiri & Gündüz, 2020b). A key metric that captures the channel degradation quality is the signal-to-noise ratio per coordinate (SNR), defined as the ratio between the average signal energy ($P$) and that of the noise ($d$), i.e. $\text{SNR} \triangleq P/d$. The larger it is the better the signal fidelity. The power budget $P$ encourages the compression of signals: if each client can transmit the same information $\boldsymbol{x}_i$ via fewer entries (smaller $d$), they can utilize more power per entry (higher SNR) and hence a more faithful signal. For the downlink communication from the server to the clients (broadcast channel), we assume that it is noiseless and thus the clients exactly receive what the server transmits (McMahan & Ramage, 2017; Konečný et al., 2016a;b). In the rest of the paper by channel we mean the uplink channel. The channel model in Eq. (1) readily generalizes to the fast fading setup as discussed in Sec. 4.

**Gradient transmission over the channel.** In the distributed optimization setting the goal is to communicate the (time-varying) local gradients $\boldsymbol{g}_i \in \mathbb{R}^d$ to the central server over the noisy channel in Eq. (1). Here we set the messages $\boldsymbol{x}_i$ as linear scaling of gradients (as we want to estimate the gradient average), i.e. $\boldsymbol{x}_i = a_i \boldsymbol{g}_i$ with the scalars $a_i \in \mathbb{R}$ enforcing the power constraints:

$$\boldsymbol{y} = \sum_{i=1}^{k} a_i \boldsymbol{g}_i + \boldsymbol{Z}, \quad \|a_i \boldsymbol{g}_i\|^2 \leq P_t. \tag{2}$$

Now the received signal is a weighted sum of the gradients corrupted by noise, whereas we need the sum of the gradients $\sum_i \boldsymbol{g}_i$ (upto zero mean additive noise) for the model training. Towards this goal, a common mild technical assumption is that the gradient norms $\{\|\boldsymbol{g}_i\|\}$ are known at the receiver at each communication round (Chang & Tandon, 2020; Guo et al., 2020) (can be relaxed in practice, Sec. 4). The optimal scalars are then given by $a_i = \sqrt{P_t}/(\max_j \|\boldsymbol{g}_j\|), \forall i \in [K]$, which are uniform across all the clients (§ E.1). Now substituting this $a_i$ in Eq. (2) and rearranging, the effective channel can be written as

$$\boldsymbol{y} = \widetilde{\mathcal{Z}}_P(\{\boldsymbol{g}_i\}) \triangleq \frac{1}{k}\sum_{i=1}^{k} \boldsymbol{g}_i + \frac{\max_i \|\boldsymbol{g}_i\|}{k\sqrt{P_t}} \boldsymbol{Z}. \qquad \text{(noisy channel)}$$

Or equivalently, we can assume this as the actual channel model where the server receives the gradient average corrupted by a zero mean noise proportional to the gradients. Note that the noise magnitude decays in time as gradients converge to zero. We denote $\widetilde{\mathcal{Z}}_P(\cdot)$ as simply $\mathcal{Z}_P(\cdot)$ henceforth as these two mappings are equivalent.

**Z-SGD.** Recall that the SGD aggregates the uncompressed gradients directly. In the presence of the noisy channel, it naturally modifies to

$$\boldsymbol{\theta}_{t+1} = \boldsymbol{\theta}_t - \gamma_t \, \mathcal{Z}_P(\{\boldsymbol{g}_i^{(t)}\}). \qquad \text{(Z-SGD)}$$

Thus Z-SGD is a canonical baseline to compare against. It has two sources of stochasticity: one stemming for the stochastic gradients and the other from the channel noise. While the gradient in the Z-SGD update still has the same conditional mean as the noiseless case (zero mean Gaussian in noisy channel), it has higher variance due to the Gaussian term. When $P = \infty$, Z-SGD reduces to SGD.

## 3 LASER: NOVEL LINEAR COMPRESSION CUM TRANSMISSION SCHEME

In this section we describe our main contribution, LASER, a novel method to compress gradients and transmit them efficiently over noisy channels. The central idea underpinning our approach is that, given the channel power constraint in Eq. (1), we can get a more faithful gradient signal at the receiver by transmitting its 'appropriate' compressed version (fewer entries sent and hence more power per entry) as opposed to sending the full-gradient naively as in Z-SGD. This raises a natural question: *what's a good compression scheme that facilitates this?* To address this, we posit that we can capitalize on the inherent low-rank structure of the gradient matrices (Martin & Mahoney, 2021; Mazumder et al., 2010; Yoshida & Miyato, 2017) for efficient gradient compression and transmission. Indeed, as illustrated below and in Theorem 1, we can get a variance reduction of the order of the smaller dimension when the gradient matrices are approximately low-rank.

More concretely, let us consider the single worker case where the goal is to transmit the stochastic gradient $g \in \mathbb{R}^{m \times m}$ (viewed as a matrix) to the server with constant power $P_t = P$. Further let's suppose that $g$ is approximately rank-one, i.e. $g \approx pq^\top$, with the factors $p, q \in \mathbb{R}^m$ known. If we transmit $g$ uncompressed over the noisy channel, as in Z-SGD, the server receives $y_{\text{Z-SGD}} = g + (\|g\|/\sqrt{P}) Z \in \mathbb{R}^{m \times m}$. On the other hand, if we capitalize on the low-rank structure of $g$ and instead transmit the factors $p$ and $q$ with power $P/2$ each, the server would receive:

$$y_p = p + (\sqrt{2}\|p\|/\sqrt{P}) Z_p \in \mathbb{R}^m, \quad y_q = q + (\sqrt{2}\|q\|/\sqrt{P}) Z_q \in \mathbb{R}^m,$$

where $Z_p$ and $Z_q$ are the channel noise. Now we reconstruct the stochastic gradient as

$$y_{\text{LASER}} \triangleq y_p y_q^\top = (p + (\sqrt{2}\|p\|/\sqrt{P}) Z_p)(q + (\sqrt{2}\|q\|/\sqrt{P}) Z_q)^\top. \quad (3)$$

Conditioned on the gradient $g$, while the received signal $y$ has the same mean $g$ under both Z-SGD and LASER, we observe that for Z-SGD it has variance $\mathbb{E}\|y_{\text{Z-SGD}} - g\|^2 = \|g\|^2/\text{SNR}$ with $\text{SNR} \triangleq P/m^2$, whereas that of LASER is roughly $\|g\|^2 \cdot (4/m\text{SNR})(1 + 1/(m\text{SNR}))$, as further elaborated in Definition 1. When SNR is of constant order $\Omega(1)$, we observe that the variance for LASER is roughly $\mathcal{O}(m)$ times smaller than that of Z-SGD, which is significant given that variance directly affects the convergence speed of stochastic-gradient based methods (Bottou et al., 2018).

More generally, even if the gradients are not inherently low-rank and we only know their rank factors approximately, with standard techniques like error-feedback (Seide et al., 2014) we can naturally generalize the aforementioned procedure, which is the basis for LASER. Algorithm 1 below details LASER and Theorem 1 establishes its theoretical justification. While LASER works with any power policy $\{P_t\}$ in noisy channel, it suffices to consider the constant law $P_t = P$ as justified in Sec. 4.2.

### 3.1 ALGORITHM

---

**Algorithm 1** LASER: Linear compression in wireless distributed optimization

---

1: **input**: initial model parameters $\theta \in \mathbb{R}^{m \times n}$, learning rate $\gamma$, compression rank $r$, power budget $P$
2: **output**: trained parameters $\theta$
3: **at** each worker $i = 1, \dots, k$ **do**
4:     **initialize** memory $e_i \leftarrow 0 \in \mathbb{R}^{m \times n}$
5:     **for** each iterate $t = 0, \dots$ **do**
6:         Compute a stochastic gradient $g_i \in \mathbb{R}^{m \times n}$
7:         $M_i \quad \leftarrow e_i + \gamma g_i$         ▷ Updated gradient via error feedback
8:         $P_i, Q_i \leftarrow \mathcal{C}_r(M_i)$         ▷ Rank-$r$ compression (Vogels et al., 2019)
9:         $e_i \quad \leftarrow M_i - \text{DECOMPRESS}(\mathcal{C}_r(M_i))$         ▷ Memorize local errors
10:         $\alpha, \beta \quad \leftarrow \text{POWERALLOC}(\{\mathcal{C}_r(M_j), M_j\})$         ▷ Power allocation
11:         $Y_p, Y_q \leftarrow \mathcal{Z}_\alpha(\{P_j\}), \mathcal{Z}_\beta(\{Q_j\})$         ▷ Channel transmission
12:         $g \quad \leftarrow \text{DECOMPRESS}(Y_p, Y_q)$         ▷ Reconstruct the gradient in $\mathbb{R}^{m \times n}$
13:         $\theta \quad \leftarrow \theta - g$
14:     **end for**
15: **end at**

---

For distributed training of neural network models, we apply Algorithm 1 to each layer independently. Further we use it only for the weight matrices (fully connected layers) and the convolutional filters (after reshaping the multi-dimensional tensors to matrices), and transmit the bias vectors uncompressed.

Now we delineate the two main components of LASER: (i) Gradient compression + Error-feedback (EF), and (ii) Power allocation + Channel transmission.

**Gradient compression and error feedback (7-9).** Since we transmit low-rank gradient approximations, we use error feedback (EF) to incorporate the previous errors into the current gradient update. This ensures convergence of SGD with biased compressed gradients (Karimireddy et al., 2019). For the rank-$r$ compression of the updated gradient $M$, $\mathcal{C}_r(M)$, we use the PowerSGD algorithm from Vogels et al. (2019), a linear compression scheme to compute the left and right singular components $P \in \mathbb{R}^{m \times r}$ and $Q \in \mathbb{R}^{n \times r}$ respectively. PowerSGD uses a single step of the subspace iteration (Stewart & Miller, 1975) with a warm start from the previous updates to compute these factors. The approximation error, $M - PQ^\top$, is then used to update the error-feedback for next iteration. Note that the clients do not have access to the channel output and only include the local compression errors into their feedback. The decompression function in line 9 is given by DECOMPRESS$(P, Q) \triangleq PQ^\top \in \mathbb{R}^{m \times n}$.

**Power allocation and channel transmission (10-11).** This block is similar to Eq. (3) we saw earlier but generalized to multiple workers and higher rank. For each client, to transmit the rank-$r$ factors $P$ and $Q$ over the noisy channel, we compute the corresponding power-allocation vectors $\alpha, \beta \in \mathbb{R}_+^r$, given by $\alpha, \beta = $ POWERALLOC$(P, Q, M)$. This allocation is uniform across all the clients. Given these power scalars, all the clients synchronously transmit the corresponding left factors over the channel which results in $Y_p \in \mathbb{R}^{m \times r}$. Similarly for $Y_q \in \mathbb{R}^{n \times r}$. Finally, the stochastic gradient for the model update is reconstructed as $g = Y_p Y_q^\top$. For brevity we defer the full details to § E.1.

### 3.2 Theoretical results

We now provide theoretical justification for LASER for learning parameters in $\mathbb{R}^{m \times n}$ with $m \leq n$ (without loss of generality). While our algorithm works for any number of clients, for the theory we consider $k = 1$ to illustrate the primary gains with our approach. Our results readily extend to the multiple clients setting following Cordonnier (2018). Specifically, Theorem 1 below highlights that the asymptotic convergence rate of LASER is *almost the same as that of the classical* SGD, except for a small additive constant $\lambda_{\text{LASER}}$ which is $\mathcal{O}(m)$ times smaller than that of Z-SGD. Our results hold for both quasi-convex and arbitrary non-convex functions. We start with the preliminaries.

**Definition 1** (**Channel influence factor**). *For any compression cum transmission algorithm* ALG, *let* $y_{\text{ALG}}(g)$ *be the reconstructed gradient at the server after transmitting $g$ over the noisy channel. Then the channel influence factor $\lambda_{\text{ALG}}$ is defined as*

$$\lambda_{\text{ALG}} \triangleq \frac{\mathbb{E}_z \|y_{\text{ALG}}(g) - g\|^2}{\|g\|^2}. \tag{4}$$

The influence factor gauges the effect of the channel on the variance of the final gradient $y_{\text{ALG}}$: if the original stochastic gradient $g$ has variance $\sigma^2$ with respect to the actual gradient $\nabla f$, then $y_{\text{ALG}}$ has $(1 + \lambda_{\text{ALG}})\sigma^2$. Note that this variance directly affects the convergence speed of the SGD and hence the smaller $\lambda_{\text{ALG}}$ is, the better the compression scheme is. In view of this, the following fact (§ B.2) illustrates the crucial gains of LASER compared to Z-SGD, which are roughly of order $\mathcal{O}(m)$:

$$\lambda_{\text{LASER}} \leq \frac{4}{(m/r)\text{SNR}} \left(1 + \frac{1}{(n/r)\text{SNR}}\right) \ll \frac{1}{\text{SNR}} = \lambda_{\text{Z-SGD}}. \tag{5}$$

In the low-rank (Vogels et al., 2019) and constant-order SNR regime where $r = \mathcal{O}(1)$ and SNR $= \Omega(1)$, we observe that $\lambda_{\text{LASER}}$ is roughly $\mathcal{O}(m)$ times smaller than $\lambda_{\text{Z-SGD}}$. In other words, the effective SNR seen by LASER roughly gets boosted to $\mathcal{O}(m\,\text{SNR})$ due to capitalizing on the low-rank factors whereas Z-SGD perceives only the standard factor SNR. Constant-order SNR, i.e. $P/mn = \Omega(1)$, means that the energy used to transmit each coordinate is roughly a constant, analogous to the constant-order bits used in quantization schemes (Vargaftik et al., 2021). In fact, a weaker condition that $P/4r^2 > 1$ suffices (§ E.3). With a slight abuse of notation, we denote the first upper bounding quantity in Eq. (5) as $\lambda_{\text{LASER}}$ too and DECOMPRESS$(\mathcal{C}_r(\cdot))$ as $\mathcal{C}_r(\cdot)$ for brevity.

We briefly recall the standard assumptions for SGD convergence following the framework in Bottou et al. (2018) and Stich & Karimireddy (2019).

**Assumption 1.** *The objective $f : \mathbb{R}^{m \times n} \to \mathbb{R}$ is differentiable and $\mu$-quasi-convex for a constant $\mu \geq 0$ with respect to $\theta_\star$, i.e. $f(\theta) - f(\theta_\star) + \frac{\mu}{2}\|\theta - \theta_\star\|^2 \leq \langle \nabla f(\theta), \theta - \theta_\star \rangle, \ \forall \theta \in \mathbb{R}^{m \times n}$.*

**Assumption 2.** *$f$ is $L$-smooth for some $L > 0$, i.e. $f(\boldsymbol{\theta}') \leq f(\boldsymbol{\theta}) + \langle \nabla f(\boldsymbol{\theta}), \boldsymbol{\theta}' - \boldsymbol{\theta} \rangle + \frac{L}{2}\|\boldsymbol{\theta}' - \boldsymbol{\theta}\|^2, \ \forall \boldsymbol{\theta}, \boldsymbol{\theta}' \in \mathbb{R}^{m \times n}$.*

**Assumption 3.** *For any $\boldsymbol{\theta}$, a gradient oracle $\boldsymbol{g}(\boldsymbol{\theta}, \boldsymbol{\xi}) = \nabla f(\boldsymbol{\theta}) + \boldsymbol{\xi}$, and conditionally independent noise $\boldsymbol{\xi}$, there exist scalars $(M, \sigma^2) \geq 0$ such that $\mathbb{E}[\boldsymbol{\xi}|\boldsymbol{\theta}] = 0$, $\mathbb{E}[\|\boldsymbol{\xi}\|^2|\boldsymbol{\theta}] \leq M\|\nabla f(\boldsymbol{\theta})\|^2 + \sigma^2$.*

**Assumption 4.** *The compressor $\mathcal{C}_r(\cdot)$ satisfies the $\delta_r$-compression property: there exists a $\delta_r \in [0, 1]$ such that $\mathbb{E}_{\mathcal{C}_r}\|\mathcal{C}_r(\boldsymbol{M}) - \boldsymbol{M}\|^2 \leq (1 - \delta_r)\|\boldsymbol{M}\|^2, \ \forall \boldsymbol{M} \in \mathbb{R}^{m \times n}$.*

$\delta_r$-compression is a standard assumption in the convergence analysis of Error Feedback SGD (EF-SGD) (Stich & Karimireddy, 2020). It ensures that the norm of the feedback memory remains bounded. We make the following assumption on the influence factor $\lambda_{\text{LASER}}$, which ensures that the overall composition of the channel and compressor mappings, $\mathcal{Z}_P(\mathcal{C}_r(\cdot))$, still behaves nicely.

**Assumption 5.** *The channel influence factor $\lambda_{\text{LASER}}$ satisfies $\lambda_{\text{LASER}} \leq 1/(10(2/\delta_r + M))$.*

We note that a similar assumption is needed for convergence even in the hypothetical ideal scenario when the clients have access to the channel output (§ B.2), which we do not have. This bound can be roughly interpreted as $\lambda_{\text{LASER}} = \mathcal{O}(\delta_r)$. We are now ready to state our main result.

**Theorem 1** (**LASER convergence**). *Let $\{\boldsymbol{\theta}_t\}_{t \geq 0}$ be the LASER iterates (Alg. 1) with constant stepsize schedule $\{\gamma_t = \gamma\}_{t \geq 0}$ and suppose Assumptions 2-5 hold. Denote $\boldsymbol{\theta}_\star \triangleq \arg\min_{\boldsymbol{\theta}} f(\boldsymbol{\theta})$, $f_\star \triangleq f(\boldsymbol{\theta}_\star)$, and $\tau \triangleq 10L\left(\frac{2}{\delta_r} + M\right)$. Then for $k = 1$,*

(i) *if $f$ is $\mu$-quasi convex for $\mu > 0$, there exists a stepsize $\gamma \leq \frac{1}{\tau(1+\lambda_{\text{LASER}})}$ such that*

$$\mathbb{E}f(\boldsymbol{\theta}_{\text{out}}) - f_\star = \widetilde{\mathcal{O}}\left(\tau(1 + \lambda_{\text{LASER}})\|\boldsymbol{\theta}_0 - \boldsymbol{\theta}^\star\|^2 \exp\left(\frac{-\mu T}{\tau(1 + \lambda_{\text{LASER}})}\right) + \frac{\sigma^2(1 + \lambda_{\text{LASER}})}{\mu T}\right),$$

*where $\boldsymbol{\theta}_{\text{out}}$ is chosen from $\{\boldsymbol{\theta}\}_{t=0}^{T-1}$ such that $\boldsymbol{\theta}_{\text{out}} = \boldsymbol{\theta}_t$ with probability $(1 - \mu\gamma/2)^{-t}$.*

(ii) *if $f$ is $\mu$-quasi convex for $\mu = 0$, there exists a stepsize $\gamma \leq \frac{1}{\tau(1+\lambda_{\text{LASER}})}$ such that*

$$\mathbb{E}f(\boldsymbol{\theta}_{\text{out}}) - f_\star = \mathcal{O}\left(\frac{\tau\|\boldsymbol{\theta}_0 - \boldsymbol{\theta}^\star\|^2(1 + \lambda_{\text{LASER}})}{T} + \sigma\|\boldsymbol{\theta} - \boldsymbol{\theta}_\star\|\sqrt{\frac{1 + \lambda_{\text{LASER}}}{T}}\right),$$

*where $\boldsymbol{\theta}_{\text{out}}$ is chosen uniformly at random from $\{\boldsymbol{\theta}\}_{t=0}^{T-1}$.*

(iii) *if $f$ is an arbitrary non-convex function, there exists a stepsize $\gamma \leq \frac{1}{\tau(1+\lambda_{\text{LASER}})}$ such that*

$$\mathbb{E}\|\nabla f(\boldsymbol{\theta}_{\text{out}})\|^2 = \mathcal{O}\left(\frac{\tau\|f(\boldsymbol{\theta}_0) - f_\star\|^2(1 + \lambda_{\text{LASER}})}{T} + \sigma\sqrt{\frac{L(f(\boldsymbol{\theta}) - f_\star)(1 + \lambda_{\text{LASER}})}{T}}\right),$$

*where $\boldsymbol{\theta}_{\text{out}}$ is chosen uniformly at random from $\{\boldsymbol{\theta}\}_{t=0}^{T-1}$.*

(iv) *Z-SGD obeys the convergence bounds (i)-(iii) with $\delta_r = 1$ and $\lambda_{\text{LASER}}$ replaced by $\lambda_{\text{Z-SGD}}$.*

**LASER vs. Z-SGD.** Thus the asymptotic rate of LASER is dictated by the timescale $(1 + \lambda_{\text{LASER}})/T$, very close to the $1/T$ rate for the classical SGD. In contrast, Z-SGD has the factor $(1 + \lambda_{\text{Z-SGD}})/T$ with $\lambda_{\text{Z-SGD}} = \mathcal{O}(m)\,\lambda_{\text{LASER}}$.

**Multiple clients.** As all the workers in LASER (Alg. 1) apply the same linear operations for gradient compression (via PowerSGD), Theorem 1 can be extended to (homogenous) multiple workers by shrinking the constants $\sigma^2$, SNR, $\lambda_{\text{LASER}}$, and $\lambda_{\text{Z-SGD}}$ by a factor of $k$, following Cordonnier (2018).

*Proof.* (Sketch) First we write the LASER iterates $\{\boldsymbol{\theta}_t\}_{t \geq 0}$ succinctly as
$$\boldsymbol{\theta}_{t+1} = \boldsymbol{\theta}_t - \mathcal{Z}(\mathcal{C}_r(\boldsymbol{e}_t + \gamma_t \boldsymbol{g}_t)),$$
$$\boldsymbol{e}_{t+1} = (\boldsymbol{e}_t + \gamma_t \boldsymbol{g}_t) - \mathcal{C}_r(\boldsymbol{e}_t + \gamma_t \boldsymbol{g}_t).$$

First we establish a bound on the gap to the optimum, $\mathbb{E}\|\boldsymbol{\theta}_{t+1} - \boldsymbol{\theta}_\star\|^2$, by the descent lemma (Lemma 11). This optimality gap depends on the behavior of the error updates via $\mathbb{E}\|\boldsymbol{e}_t\|^2$, which we characterize by the error-control lemma (Lemma 12). When $f$ is quasi-convex, these two lemmas help us establish a recursive inequality between the optimality gap $\mathbb{E}f(\boldsymbol{\theta}_{t+1}) - f_\star$ at time $t + 1$ and with that of at time $t$: $\mathbb{E}f(\boldsymbol{\theta}_t) - f_\star$. Upon unrolling this recursion and taking a weighted summation, Lemma 3 establishes the desired result. In the case of non-convexity, the same idea helps us to control $\mathbb{E}\|\nabla f(\boldsymbol{\theta}_t)\|^2$ in a similar fashion and when combined with Lemma 6, yields the final result. The proof for Z-SGD is similar. $\qquad\square$

**Table 2:** Benchmarks for evaluating LASER. Baseline refers to the noiseless SGD.

| Task | Model | Params | Dataset | Metric | Baseline |
|------|-------|--------|---------|--------|----------|
| Language modeling | GPT-2 | 123.6 M | WIKITEXT-103 | Perplexity | 19.2 |
| Image classification | RESNET18 | 11.2 M
11.2 M | CIFAR10
CIFAR100 | Top-1
accuracy | 93.0%
73.1% |
| | 1-LAYER NN | 7850 | MNIST | | 92.3% |

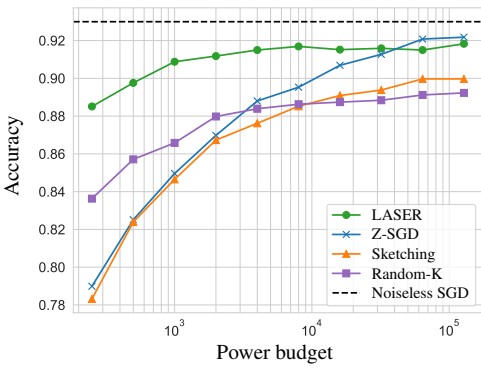

**Figure 2:** Test accuracy (*higher the better*) for a given power budget on CIFAR-10 for different algorithms. LASER demonstrates consistent accuracy gains over the baselines over a wide range of power levels.

**Table 3:** Power required (*lower the better*) to reach the given target accuracy on CIFAR-10. LASER requires 16× lesser power than the Z-SGD to achieve the same targetaccuracy. Equivalently, LASER tolerates more channel noise than the Z-SGD for the same target accuracy as is partly supported by our theoretical analysis.

| Target | Power required | | Reduction |
|--------|-------|-------|-----------|
| | LASER | Z-SGD | |
| 88% | 250 | 4000 | 16× |
| 89% | 500 | 8000 | 16× |
| 90% | 1000 | 16000 | 16× |
| 91% | 2000 | 32000 | 16× |

## 4 EXPERIMENTAL RESULTS

We empirically demonstrate the superiority of LASER over state-of-the-art baselines on a variety of benchmarks, summarized in Table 2.

**Setup.** We consider four challenging tasks of practical interest: (i) GPT language modeling on WIKITEXT-103, and (ii, iii, iv) image classification on MNIST, CIFAR10 and CIFAR100. For the language modeling, we use the GPT-2 like architecture following Pagliardini (2023) (§ F). RESNET18 is used for the CIFAR datasets. For MNIST, we use a 1-hidden-layer network for a fair comparison with Amiri & Gündüz (2020b). For distributed training of these models, we consider $k = 4$ clients for language modeling and $k = 16$ for image classification. We simulate the noisy channel by sampling $\boldsymbol{Z} \sim \mathcal{N}(0, \mathbf{I}_d)$. To gauge the performance of algorithms over a wide range of noisy conditions, we vary the power $P$ geometrically in the range $[0.1, 10]$ for MNIST, $[250, 128000]$ for CIFAR10 and CIFAR100, and $[10000, 1024 \times 10000]$ for WIKITEXT-103. The chosen ranges can be roughly split into low-moderate-high power regimes. Recall from noisy channel that the smaller the power, the higher the noise in the channel.

**Baselines.** We benchmark LASER against three different sets of baselines: (i) Z-SGD, (ii) SIGNUM, RANDOM-K, SKETCHING, and (iii) A-DSGD. Z-SGD sends the uncompressed gradients directly over the noisy channel and acts as a canonical baseline. The algorithms in (ii) are state-of-the-art distributed compression schemes for noiseless communication (Vogels et al., 2019). SIGNUM (Bernstein et al., 2018) transmits the gradient sign followed by the majority vote and SKETCHING (Rothchild et al., 2020; Haddadpour et al., 2020) uses a Count Mean Sketch to compress the gradients. We omit comparison with quantization methods (Vargaftik et al., 2022) given the difference in our objectives and the settings (noisy channel). A-DSGD (Amiri & Gündüz, 2020b) is a popular compression scheme for noisy channels, relying on Top-K and random sketching. However A-DSGD does not scale to tasks of the size we consider and hence we benchmark against it only on MNIST. SGD serves as the noiseless baseline (Table 2). All the compression algorithms use the error-feedback, and use the compression factor (compressed-gradient-size/original-size) 0.2, the optimal in the range $[0.1, 0.8]$. We report the best results among 3 independent runs for all the baselines (§ F).

**Table 4:** Test accuracy (*higher the better*) after 50 epochs on MNIST for low, moderate, and high power regimes.

| Algorithm | Test accuracy | | |
|---|---|---|---|
| | $P = 0.1$ | $P = 1$ | $P = 10$ |
| Z-SGD | 81.3% | 87.9% | 91.9% |
| SIGNUM | 76.7% | 83.2% | 85.4% |
| RANDOM-K | **86.1**% | 89.3% | 91.5% |
| SKETCHING | 81.9% | 88.2% | 91.7% |
| A-DSGD | 81.6% | 86.9% | 87.3% |
| LASER | 84.3% | **89.9**% | **92.3**% |

**Table 5:** Communication cost (*lower the better*) for GPT language modeling on WIKITEXT-103. LASER transmits the lowest volume of data during training.

| Algorithm | Data sent per iteration | |
|---|---|---|
| Z-SGD | 496 MB | (1×) |
| SIGNUM | 15 MB | (33×) |
| RANDOM-K | 99 MB | (5×) |
| SKETCHING | 99 MB | (5×) |
| A-DSGD | n/a | n/a |
| LASER | **3 MB** | (165×) |

## 4.1 RESULTS ON LANGUAGE MODELING AND IMAGE CLASSIFICATION

For GPT language modeling, Fig. 1 in Sec. 1 highlights that LASER outperforms the baselines over a wide range of power levels. To the best of our knowledge, this is the first result of its kind to demonstrate gains for GPT training over noisy channels. Specifically, we obtain $64\%$ improvement in perplexity over Z-SGD (76 vs. 212) in the low power regime ($P = 10\,\text{K}$) and $50\%$ (35 vs. 71) for the moderate one ($P = 160\,\text{K}$). This demonstrates the efficacy of LASER especially in the limited power environment. Indeed, Table 1 illustrates that for a fixed target perplexity, LASER requires $16\times$ less power than the second best, Z-SGD. In the very high power regime, we observe no clear gains (as expected) compared to transmitting the uncompressed gradients directly via the Z-SGD.

We observe a similar trend for CIFAR10 classification, as Fig. 2 and Table 3 demonstrate the superiority of LASER over other compression schemes; RANDOM-K does better than the other baselines till moderate power levels after which Z-SGD dominates. SIGNUM is considerably worse than others, as it hasn't converged yet after 150 epochs, and hence omitted. With regards to power reduction, Table 3 highlights that LASER requires just $(1/16)^{\text{th}}$ the power compared to Z-SGD to reach any target accuracy till $91\%$. We observe similar gains for CIFAR100 (§ F).

Table 4 compares the performance of LASER against various compression algorithms on MNIST. In the very noisy regime ($P = 0.1$), RANDOM-K is slightly better than LASER and outperforms the other baselines, whereas in the moderate ($P = 1$) and high power ($P = 10$) regimes, LASER is slightly better than the other algorithms. On the other hand, we observe that A-DSGD performs worse than even simple compression schemes like RANDOM-K in all the settings.

## 4.2 POWER CONTROL: STATIC VS. DYNAMIC POLICIES

The formulation in noisy channel allows for any power control law $P_t$ as long as it satisfies the average power constraint: $\sum_t (P_t/T) \leq P$. This begs a natural question: *what's the best power scheme for LASER?* To answer this, for CIFAR10 classification, under a fixed budget $P$ we consider different power policies with both increasing and decreasing power across epochs: the constant, piecewise constant and linear schemes. Fig. 3 illustrates the results for the decreasing power laws, while Fig. 7 their increasing counterparts. These results highlight that the *constant* power policy achieves the *best* performance for both LASER and Z-SGD, compared to the time-varying ones. Further LASER attains significant accuracy gains over Z-SGD for all the power control laws. Interestingly LASER performs

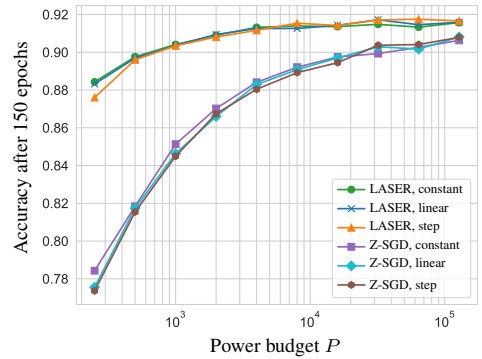

**Figure 3:** Accuracy vs. budget $P$ for various laws. Constant is the best for both LASER and Z-SGD.

the *same* with all the power schemes. We posit this behavior to the fact that the noisy channel already contains a time-varying noise due to the term $\frac{\max_i \|\|\boldsymbol{g}_i\|\|}{\sqrt{P_t}}$. Since the gradients decay over time, this

inherently allows for an implicit power/SNR-control law even with a constant $P_t$, thus enabling the constant power scheme to fare as good as the others. Hence, without loss of generality, we consider the static power schedule for our theory and experiments. We refer to § F.7 for a detailed discussion.

### 4.3 COMPUTATIONAL COMPLEXITY AND COMMUNICATION COST

Recall from Algorithm 1 that the two critical components of LASER are gradient compression and channel transmission. To gauge their efficacy we analyze them via two important metrics: (i) *computational complexity* of compression and (ii) *communication cost* of transmission. For (ii), recall from Eq. (1) that the power constraint indirectly serves as a communication cost and encourages compression. Table 5 quantitatively measures the total data sent by clients for each training iteration (doesn't change with the power $P$) for GPT language modeling on WIKITEXT-103. As illustrated, LASER incurs the lowest communication cost among all the baselines with $165\times$ cost reduction as compared to the Z-SGD, followed by SIGNUM which obtains $33\times$ reduction. Interestingly, LASER also achieves the best perplexity scores as highlighted in Fig. 1. For these experiments, we let rank $r = 4$ for LASER and the best compression factor $0.2$ for the baselines (as detailed earlier). SIGNUM does not require any compression factor. For (i), since LASER relies on PowerSGD for the rank decomposition, it inherits the same low-complexity benefits: Tables 3-7 of Vogels et al. (2019) demonstrate that PowerSGD is efficient with significantly lower computational needs and has much smaller processing time/batch as compared to baselines without any accuracy drop. In fact, it is the core distributed algorithm behind the recent breakthrough DALL-E (§ E in Ramesh et al. (2021)).

**Slow and fast fading channels.** The slow/non-fading model in Eq. (1) readily generalizes to the popular fast fading channel (Guo et al., 2020; Amiri & Gündüz, 2020a): $\boldsymbol{y} = \sum_i \gamma_i \boldsymbol{x}_i + \boldsymbol{Z}$, where $\gamma_i$ are the channel fading coefficients. A standard technique here in the literature is to assume that channel-state-information (CSI) is known in the form of fading coefficients or their statistics, which essentially reduces the problem to a non-fading one. Likewise LASER can be extended to the fast fading channel as well. The challenging setting without CSI is an interesting topic of future research.

## 5 RELATED WORK

**(i) Compression schemes with noiseless communication.** Assuming a noiseless bit pipe from clients to the server, quantization methods (Dettmers, 2015; Alistarh et al., 2017; Horvóth et al., 2022; Li et al., 2018; Wen et al., 2017; Yu et al., 2019; Vargaftik et al., 2021) quantize each coordinate and send as fewer bits as possible. Sparsification techniques (Ivkin et al., 2019; Stich et al., 2018; Sun et al., 2019; Tsuzuku et al., 2018; Wangni et al., 2018) send a reduced number of coordinates, based on criteria such as Top/Random-K, as opposed to sending the full gradient directly. Hybrid methods (Dryden et al., 2016; Lim et al., 2019) combine both. Rank compression methods (Yu et al., 2018; Cho et al., 2019; Wang et al., 2018) spectrally decompose gradient matrix (often via SVD) and transmit these factors. Since SVD is computationally prohibitive, we rely on the state-of-the-art light-weight compressor PowerSGD (Vogels et al., 2019). **(ii) Compression schemes for noisy channels.** The main idea here is to enable over-the-air-aggregation of gradients via the superposition nature of wireless channels (Nazer & Gastpar, 2007) thus reducing the communication latency and bandwidth. The popular A-DSGD (Amiri & Gündüz, 2020b) relies on Top-K sparsification and random sketching. However, being memory intensive, A-DSGD is restricted to MNIST with 1-layer NN and doesn't scale beyond. Guo et al. (2020) propose an analog-gradient-aggregation scheme but it is limited to shallow neural networks. Chang & Tandon (2020) design a digital quantizer for training over Gaussian MAC channels. **(iii) Power laws.** In the absence of explicit power constraints, Wei & Shen (2022a) show that $\mathcal{O}(1/t^2)$ noise-decay ensures the standard $1/T$ convergence rate for noisy FED-AVG whereas Saha et al. (2022) propose a $t^{0.8}$ increase in SNR for the decentralized setup.

## 6 CONCLUSION

We propose a principled gradient compression scheme, LASER, for wireless distributed optimization over additive noise channels. LASER attains significant gains over its baselines on a variety of metrics such as accuracy/perplexity, complexity and communication cost. It is an interesting avenue of future research to extend LASER to channels with downlink noise and fast fading without CSI.

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
