# OpenReview forum: "LASER: Linear Compression in Wireless Distributed Optimization"
_ICLR.cc/2024/Conference — Submitted to ICLR 2024_

### Official Review · Reviewer_dVrm · 2023-10-31

**Soundness:** 3 good
**Presentation:** 3 good
**Contribution:** 2 fair
**Rating:** 6
**Confidence:** 4

**Summary:**

This paper  introduces the LASER scheme, a novel communication-efficient distributed optimization approach utilizing plain SGD. In contrast to most existing literature, LASER incorporates considerations for communication noise. The compression method implemented within LASER involves both low-rank representation and gradient scaling, with a detailed algorithmic description provided. Theoretical analyses are included to demonstrate the convergence assurance of the proposed scheme, while experimental evaluations validate the complexity improvements.

**Strengths:**

1. The algorithm design considers the noise in communication, which is more realistic than most works in literature. The design enables the level of noise to decrease when the norm gradient is smaller. This is important to guarantee good performance of SGD.
2. The convergence analysis of the proposed scheme is comprehensive and the result is reasonable. The convergence rate in quansi-convex and non-convex setting is comparable to literature and standard distributed optimization methods.
2. The experiments setup is closely related to the theoretical analysis and the result is convincing.

**Weaknesses:**

1. The major concern is the novelty of this work. The low-rank approximation of gradient or weight matrix is a common method in literature. And this method has been applied in distributed learning, especially Federated learning setting, e.g, Zhou, Huachi, et al. "Low rank communication for federated learning." , 2020,Konečný, Jakub, et al. "Federated learning: Strategies for improving communication efficiency." arXiv preprint arXiv:1610.05492 (2016). Thus, I do not think the proposed framework if of great contribution.
2. The power allocation step is also not novel. The method introduced in equation (2) is a standard noise variance minimization method in 'Guo, Huayan, An Liu, and Vincent KN Lau. "Analog gradient aggregation for federated learning over wireless networks: Customized design and convergence analysis." IEEE Internet of Things Journal 8.1 (2020): 197-210.'
3. The theoretical analysis is based on standard distributed optimization with error feedback. The only difference is the low-rank compression and noise are considered.

**Questions:**

1. Looks like LASER is a combination of well-known methods(low-rank approximation and noise variance minimization), so is there any major difference or difficulty in theoretical analysis besides the ones are already sound in literature?
2. Could you provide intuitive understanding about why low-rank approximation can induce lower noise variance? Specifically, explain the intuition on the comparison of noise variance in equation (5).

---

> ### Author Response · Authors · 2023-11-17
> **Novelty and theoretical contributions**
>
> We thank the reviewer for the constructive feedback and insightful comments. We refer to the common response about the  theoretical and  algorithmic novelty of LASER.
>
> - **Theoretical analysis:** As highlighted in the common response, a major difficulty in the theoretical analysis of LASER is handling the communication noise and the power budget. Since the channel noise interacts in a non-linear way with the gradients (Eq. (noisy channel)), we tackle this through introducing **channel-influence** factor which gives a handle to control the second-moments of the noisy gradients, which are crucial for convergence analysis. On the power front, Lemmas 7 and 8 in Appendix B.1 establish the optimal power allocation scheme among the rank components in order to obtain a tight characterization of the channel influence for various compression schemes. This further facilitates a thorough comparison with baselines such as Z-SGD and showcases the benefits of compression algorithms through convergence rates.
>
> - **Intuition behind Eq. (5):** As demonstrated in Section 3 of the paper, the underlying intuition here is that under a fixed power budget, if we transmit the full gradient matrix of size $m \times m$, this budget has to be utilized for $m^2$ entries. On the other hand, if we capitalize on its low-rank structure we effectively need to transmit $O(m)$ entries, thus we can allocate more power-per-entry hence resulting in a roughly $O(m)$ boost in the signal-to-noise ratio (SNR). This intuition mathematically translates into the reduction of the gradient variance, as captured by the channel-influence factor in Eq. (4). In view of this, Eq. (5) compares these noise variances for Z-SGD and LASER and confirms the above intuition.

---

> ### Author Response · Authors · 2023-11-23
> **Thank you!**
>
> We thank the reviewer for the constructive feedback and encouraging comments and reconsideration of the score!

---

### Official Review · Reviewer_DPBP · 2023-11-06

**Soundness:** 2 fair
**Presentation:** 2 fair
**Contribution:** 2 fair
**Rating:** 6
**Confidence:** 3

**Summary:**

The authors address the communication bottleneck issue in data-parallel SGD, which is widely used for distributed optimization in large-scale machine learning. The authors highlight that existing compression schemes either assume noiseless communication links or fail to perform well in practical tasks. To bridge this gap, the authors propose LASER, a gradient compression scheme that transmits gradients over noisy channels by leveraging the inherent low-rank structure.

**Strengths:**

The authors demonstrate that LASER consistently outperforms baseline methods across various benchmarks, outperforming state-of-the-art compression schemes in computer vision and GPT language modeling tasks.

**Weaknesses:**

The novelty of the paper is not well stated. It is unclear whether low-rank matrix decomposition techniques already exist and if the contribution of the paper lies solely in the utilization of low-rank decomposition to reduce gradient transmission traffic.

The practicality of the compression method is questionable. Adopting low-rank decomposition on gradients requires massive computations. It is unclear whether this operation introduces additional overhead to the distributed training procedure.

**Questions:**

Please see the weakness part.

---

> ### Author Response · Authors · 2023-11-17
> **Novelty of LASER and efficiency of its rank decomposition**
>
> We thank the reviewer for the constructive feedback and insightful comments. We refer to the common response about the theoretical and algorithmic novelty of LASER, which also illustrates that it's extremely efficient and practical.

---

### Official Review · Reviewer_t6dq · 2023-11-07

**Soundness:** 2 fair
**Presentation:** 3 good
**Contribution:** 2 fair
**Rating:** 6
**Confidence:** 4

**Summary:**

Considering the randomness in the communication environment, to reduce the power needs of sending information from the local client, the author proposed a new distributed algorithm called LASER. Different from the previous algorithm, the proposed algorithm uses low-rank compression to reduce the vector dimension that will be sent to the server. Combined with error feedback, the algorithm's convergence is established in quasi-convex, convex, and nonconvex cases. The experiments show the proposed algorithm can achieve a similar performance with lower power.

**Strengths:**

1. Consider the power that is needed to transmit vectors from the client to the server.

2. To use the power efficiently, the authors introduce a low-rank compressor to make the SNR larger than transmitting the full matrix.

3. To eliminate the error from the compressors, the authors introduce the error feedback into the proposed algorithm.

4. The authors show the convergence of the proposed algorithm under quasi-convex, convex, and non-convex cases.

5  The experimental results show that the proposed algorithm can achieve a similar performance but requires less power.

**Weaknesses:**

1. In the distributed setting, how can each client know $max ||g||$?

2. The low-rank compressor seems to be time-consuming. Why not use top-k or other compressors?

3. It seems to be unfair to compare the energy budget in each iteration, because adding a compressor even with the error feedback, the algorithm needs more iterations to converge. Thus, it would be fair to compare total energy costs. Otherwise, the most energy-saving algorithm in this framework will be the top-1 compressor (sending only one value each time).

**Questions:**

See weakness.

---

> ### Author Response · Authors · 2023-11-17
> **Processing time for PowerSGD**
>
> We thank the reviewer for the encouraging feedback and insightful comments.
>
> - **Power scalars:**  As highlighted in [1, 2], in each communication round the nodes first transmit the gradient norms $\|\| \boldsymbol{g}_i \|\|$, which are scalars, to the server which then computes their maximum and transmits it back. Since this procedure only involves transmission of real scalars, it has negligible communication overhead as compared to the gradients being transmitted.
>
> - **Processing time for PowerSGD and baselines:** As highlighted in Section 4.3 of the paper, PowerSGD [3], the low-rank compressor behind LASER, takes significantly smaller processing time as compared to other compressors such as Top-K or Random-K. For example, Table 4 in [3] highlights that PowerSGD achieves a significant 42 % reduction in processing time/batch as compared to Top-K, whereas it’s 55 % compared to Random-K for CIFAR-10. Similar computational gains of PowerSGD over the baselines are demonstrated further in [3] over various challenging datasets.
>
> - **Energy budget:** Indeed, our current comparison already considers the total energy costs in an implicit manner. Since we fix the power budget per each iteration across various baselines and compare the final accuracy after a specific number of epochs, this is equivalent to the former scenario. We observe similar gains for LASER even when the final number of epochs is increased further.
>
> **References:**
>
>
> - [1] Huayan Guo, An Liu, and Vincent KN Lau. Analog gradient aggregation for federated learning over wireless networks: Customized design and convergence analysis. *IEEE Internet of Things Journal*, 8(1):197–210, 2020.
>
> - [2] Wei-Ting Chang and Ravi Tandon. Mac aware quantization for distributed gradient descent. *In GLOBECOM 2020-2020 IEEE Global Communications Conference*, pages 1–6. IEEE, 2020.
>
> - [3]  Thijs Vogels, Sai Praneeth Karimireddy, and Martin Jaggi. PowerSGD: Practical low-rank gradient compression for distributed optimization. *Advances in Neural Information Processing Systems*, 32, 2019.

---

> > ### Comment · Reviewer_t6dq · 2023-11-22
> >
> > Thanks for the author's response, my concerns are well addressed.

---

> > > ### Author Response · Authors · 2023-11-23
> > > **Thank you!**
> > >
> > > We thank the reviewer for the constructive feedback and encouraging comments!

---

### Official Review · Reviewer_ykHv · 2023-11-09

**Soundness:** 3 good
**Presentation:** 3 good
**Contribution:** 3 good
**Rating:** 6
**Confidence:** 4

**Summary:**

The paper presents a low-rank compression method for distributed optimization. The emphasis is on wireless communication systems where the averaging is done "over the air" in a noisy channel. The performance is evaluated in experiments with both language modeling and image classification tasks.

**Strengths:**

- The incorporation of a power budget and power allocation in the compression algorithm is somewhat new and interesting.
- The consideration of a GPT model in the experiments is good.

**Weaknesses:**

- The considered wireless communication system with "over the air" averaging is essentially doing the averaging in the analog domain. This is impractical as all modern wireless systems are digital and data transmission involves channel coding, which is likely incompatible with "over the air" averaging. I am aware that this concept has been presented in many papers published in wireless communications journals and conferences, but it is still impractical. It is unlikely that wireless communication standards will incorporate a specific physical layer technique only for the sake of computing averages, which in any foreseeable future represents only a very small fraction of data transmitted in common wireless systems. Moreover, this mechanism restricts the averaging to clients communicating with a single basestation, while in practice, the coverage of a single basestation is quite small and having all the clients connecting to the same basestation is very unlikely.
- The paper focuses on low-rank compression, which by itself is a known technique. Yet, the result in Theorem 1 and its assumptions seem to be a standard error-feedback result. In particular, Assumption 4 is standard and does not capture anything specific to the fact of using low-rank compression instead of other compression methods such as top-k and random-k. It seems the only difference from standard error-feedback compression analysis is the noise term, which, however, is a straightforward extension.
- It is not quite clear how power reduction is achieved as shown in the experiments. The power budget is fixed according to Algorithm 1. If there is a fixed power budget, the same budget should apply to both the proposed algorithm and the baselines. The results in Table 1 and Table 3 either do not enforce a fixed power budget, which contradicts with the description in Algorithm 1, or there is something else wrong.

**Questions:**

Please refer to the weaknesses.

In addition, the ICLR template suggests that the appendix should be included at the end of the same PDF as the main paper.

---

> ### Author Response · Authors · 2023-11-18
> **Compatibility with Wireless Communication Standards**
>
> We thank the reviewer for the engaging and insightful comments.
>
> It is true that in today’s communication systems, the encoders need to follow standards. Related to that, we (1) explain how we could make LASER compatible with the current modulation scheme, (2) discuss potential (minimal) changes in the wireless systems required to implement LASER, and (3) make final remarks on our thoughts on academic research.
>
> **1) Digital vs. Analog transmissions:** LASER transmission as a high-order QAM.
> There are ways to adapt LASER to a digital transmission framework if needed. Perhaps one straightforward way would be to quantize the transmitted values of LASER to make it compatible with the modulation schemes used in today’s communication systems (e.g., QAM). The typical range of QAM levels varies from system to system. For WiFi, a very high modulation order (e.g., 4096-QAM) is one of the key enhancements of WiFi 7 (IEEE 802.11be). It is shown in the literature that the quantization of real-valued codewords to 6 bits usually does not affect the reliability much. The quantification of LASER is an interesting future research topic.
>
> **2) LASER and Comm. Standards.**
>
> Small modification: We would like first to note that using LASER (+ quantization) in today’s wireless systems will only require ‘bypassing’ channel coding/modulation and directly generating modulation symbols, which is not a complicated modification.
>
> LASER as a joint-source channel coding algorithm: regarding the reviewer’s comment that “Wireless communication standards will (not) incorporate a specific physical layer technique only for the sake of computing averages”, on the one hand, we’ll note that given that we do not need major changes to the wireless system as noted above, we do not think LASER is completely ruled out solely for the compatibility reason. On the other hand, we would like to note that LASER belongs to a broad family of joint source-channel coding algorithms from the perspective of which part of the standards needs to change. Communication systems, to date, relied on source-channel separation for several reasons; it’s simple; it’s known to be optimal if we have a lot of (asymptotically many) IID sources; and there are not many good joint source-channel coding algorithms. With the advance of ML, there are several reasons to explore neural joint source-channel codecs, some of which includes that not all part of the data source is equally important; we finally have the capability to design analytically intractable joint source-channel coding algorithms in a data-driven manner, etc. (cf. There are several studies demonstrating the efficacy of data-driven short-blocklength neural codecs and semantic communications such as NECST [1]). Joint source-channel coding, also called semantic communications, for wireless is definitely of interest to industry and academia; the communities as a whole are at the stage of exploring what is feasible and how much gain we can achieve from it.
>
> Recent trends in Wireless: Given the extensive interest from industry and academia in neural-based codecs, joint source-channel coding, ML-driven wireless, wireless for ML applications, end-to-end ML-driven communication blocks, and virtualized RAN, we do not think LASER (and more generally directly mapping data to modulation symbols) is completely ruled out in future. An interesting discussion would be on which communication modes (cellular, wifi, or private networks) can benefit most by deploying LASER.
>
>
> Final remark: As a final remark, we believe, as academics, we should not bind our research only to what is available with today’s technology. We believe it's an important and exciting area of research to evaluate the capability of aggregation-leveraging communication algorithms.
>
> **References:**
> - [1] NECST: https://arxiv.org/abs/1811.07557

---

> ### Author Response · Authors · 2023-11-18
> **Base station**
>
> **1)** Please note that the central node does not have to be a base station; the central node could be the edge device, such as a router in a concert hall.  As a concrete example, let's consider the following scenario.
>
> Location: a concert hall with a few hundred to several thousands of audience (or more)
>
> Clients: cell phone
>
> Central node: edge device
>
> ML models to be learned: music-related neural network in an app (e.g., Spotify)
>
> Here, several clients - whose data will be most relevant to training a music-related ML model- are physically co–located and thus naturally share the wireless medium. (Cf. The clients here would be anyone willing to share their gradients and need not be chosen beforehand.)  Thus in this example, the central node could be the edge device, such as a router in the concert hall.
>
> **2)** Applying our algorithm to “cellular” communication scenarios (i.e., Base Stations = central node, User Equipment (UE) = clients ) will potentially require more coordination and adjustments. First of all, we agree that not all cells would be suitable to perform LASER-based federated learning. Instead, only some base stations (with sufficiently large clients participating in the federated learning) will allocate some (wireless) resource blocks to the LASER operation. There are several other aspects to coordinate such as resource allocation (data traffic vs. federated learning traffic). These are all interesting potential research problems although they are not within the scope of our work. We would like to briefly mention that supporting ML applications from wireless is of increasing interest.

---

### Official Review · Reviewer_yf4L · 2023-11-10

**Soundness:** 4 excellent
**Presentation:** 4 excellent
**Contribution:** 4 excellent
**Rating:** 8
**Confidence:** 3

**Summary:**

The authors present a novel technique for distributed optimization: LASER, which is mostly composed of a low-rank compression step, and an over-the-air aggregation step. The authors present the theoretical advantage of such method over vanilla over-the-air SGD, and the convergence of the method is ensured. Finally, extensive experiments show the advantage of the proposed method, including on large scale tasks such as GPT language modeling tasks.

**Strengths:**

## Originality:

The work is original, as it is the first one to study over-the-air gradient aggregation using a low-rank compression of the gradients. The results proven regarding the channel influence factor are novel and seem not trivial to show. The experiments show an improved performance over state of the art algorithms.

## Quality:

The quality is good, with Theorems and their assumption clearly stated. Detailed proofs are provided for the results exposed. Additionally, the code is provided in the Supplementary material.

## Clarity:

I believe the contributions are clear, and well organized.

## Significance:

I believe the work is significant, in particular given the extensive experimental comparison with state of the art algorithms, including a very encouraging experimental results on large scale tasks such as GPT language modeling, which have become prominent in machine learning. Additionally, the results proven regarding the channel influence factor seem non-trivial and therefore should be very useful for the research community to build upon.

**Weaknesses:**

I just have a few questions, see “Questions” below.

**Questions:**

I just have a few questions below:

1. In algorithm 1, the local error $e_i$ is only $\boldsymbol{M}_i - \text{DECOMPRESS}(\mathcal{C}_r(\boldsymbol{M}_i))$, without ever being multiplied by $\gamma$: is this correct ? It seems that if so, the error may not be compensated correctly ? (I may be mistaken)
2. If I understand correctly, Assumption 5 is a new assumption introduced in the paper: unless I missed it in the paper, I think it would be good to elaborate a bit further on why such assumption should be verified in the settings considered, theoretically and/or experimentally (or even just with a discussion, just to provide some intuition on why such assumption should be verified): it seems that to verify it, either the compression should be very accurate (i.e. $\delta_r$ large), or $\lambda_{\text{LASER}}$ should be small, but however, while taking a larger $r$ would make $\delta_r$ larger (more accurate compression), it would also make the bound on $\lambda_{\text{LASER}}$ larger, according to eq. (5), therefore it seems a bit unclear whether Assumption 5 can be verified (or how to make it verified in practice).
3. For the experiments, I believe it would be useful to just recall how one can deal with parameters of DNNs, which are not, per say, 1 matrix: looking into Vogels 2019, it seems that, to do that, the low-rank decompositions are done layer-wise (since each layer can be seen as a matrix, even in the convolutional case), but just a quick recall about this would be useful.
4. Minor remarks/typos:
- in 5. “decompose gradient matrix” -> “decompose the gradient matrix”
- In the supplemental, just before D.2: “experssion” —> “epxression”

---

> ### Author Response · Authors · 2023-11-17
> **Intuition behind Assumption 5**
>
> We thank the reviewer for the encouraging feedback and valuable suggestions. We will fix the typos in our revised version. We address the individual questions below.
>
> **Local error:** Following the standard convention in [1], in Line 7 of Algorithm 1 we add the local error $ \boldsymbol{e}_i $ to the scaled gradient $\gamma  \boldsymbol{g}_i $  to get the new gradient $  \boldsymbol{M}_i $. In Line 13, after its low-rank reconstruction through $ \boldsymbol{g}$, a normal SGD step is performed on $ \boldsymbol{\theta}$. We can see that when there is no compression error, $ \boldsymbol{e}_i =0 $, this falls back to the classical SGD. As suggested, one could in principle multiply the learning rate $\gamma$ to the local-error but this requires a slight change in the definition of local error and the SGD descent step, so that we could recover the classical SGD in the special-case as above.
>
> **Assumption 5:** Thanks for your insightful comment. We indeed highlight the interplay between $\lambda_{\mathrm{LASER}}$ and $\delta_r$ in Appendix B.2. Since the channel corruption follows the rank-compression, Assumption 5 roughly ensures that the overall compression is still benign. Mathematically, as derived in B.2, even in the hypothetical scenario where the compressor has access to the channel output, we see that a condition of the form $\lambda_{\mathrm{LASER}} \leq \delta_r$ is needed to ensure that the overall compression factor is still less than $1$. In view of this, Assumption 5 can be roughly treated as $\lambda_{\mathrm{LASER}} \leq O(\delta_r)$ in the general scenario.
>
> On the other hand, while the above assumption is needed only for theoretical analysis, empirically we observe an interesting rank-accuracy tradeoff as demonstrated in Appendix F.5. We observe that either with low-rank or high-rank decomposition, the final accuracy is worse than that of the medium-rank. We believe this phenomenon is perhaps an empirical manifestation of the interplay between  $\lambda_{\mathrm{LASER}}$ and $ \delta_r$. Given the difficulty in characterizing $\delta_r$ precisely, establishing a clear mathematical justification for the aforementioned phenomena is an interesting topic of future research.
>
> **Layer-wise decomposition:** We fully agree with your comment. Indeed, we already specify this feature just below Algorithm 1 on page 4 in the paper, but we will further make it clear.
>
> **References:**
>
> - [1] Sebastian U Stich and Sai Praneeth Karimireddy. The error-feedback framework: Better rates for SGD with delayed gradients and compressed communication. arXiv preprint arXiv:1909.05350, 2019.

---

> ### Comment · Reviewer_yf4L · 2023-11-22
> **Response to Authors**
>
> Dear Authors, thanks a lot for your answer,
>
> - **Local error:** That makes sense, thanks for your answer (also, this has become clearer for me upon inspection of equation (20) in [1]).
> - **Assumption 5:** Thank you, that makes it clearer for me. Though, regarding the bound on $\delta\_r$, in general, wouldn't $\delta\_r$ be equal to $\frac{r}{d}$ ? (since, assuming the worst case scenario where $\boldsymbol{M}$ has all singular values equal to its largest ($\sigma\_{max}$), we have for the Frobenius norm $\\| \mathcal{C}\_r (\boldsymbol{M}) -\boldsymbol{M} \\|\_{F}^2 = (d-r)\sigma\_{max}^2 = (1 - \frac{r}{d}) d \sigma\_{max}^2 =  (1 - \frac{r}{d}) \\| \boldsymbol{M} \\|\_F^2 $ (I may be mistaken). With such bound, this means, according to equation (5), that it would be necessary to enforce an SNR large enough to ensure validity of Assumption 5 (e.g. it is not verified if SNR=1) ? However, I guess this bound on $\delta\_r$ is loose in practice since DNNs for instance, are low-rank compressible so $\delta\_r$ will be smaller. But I guess it could be interesting to deepen this aspect: for instance if there are some common assumptions on the "low-rank compressiveness" of DNNs in the literature which would give some tight enough bound on $\delta\_r$, those would in turn result in the necessary value for the SNR; another option could be to just verify experimentally, by computing the maximum $\delta\_r$ over all $\boldsymbol{M}$ encountered during training, and checking whether it verifies Assumption 5.  In any case, such analysis does not need to be done (as the remaining time is limited), and so this will not impact my evaluation of the paper, I am just checking whether my understanding of the technique is correct.
> - **Layer-wise decomposition:** Thank you, I had missed the reference on page 4.
>
> I have read the reviews and responses, and I keep my positive impression on the paper and will preserve my score.

---

> > ### Author Response · Authors · 2023-11-23
> > **Thank you!**
> >
> > We thank the reviewer for the encouraging comments about the paper. Regarding Assumption 5, had the compression $\mathcal{C}_r$ been an exact rank-$r$ compression, $\delta_r$ would have been $r/d$ as you noted. However, since we use PowerSGD for this rank-$r$ compression, it's only an approximation for the same and given the fact it uses power iteration based algorithm to compute the low-rank factors, it's a bit hard to theoretically characterize $\delta_r$. Nonetheless, as you have suggested, experimentally computing this quantity and verifying it in the context of DNNs is indeed a cool idea and worth exploring further.

---

> > > ### Comment · Reviewer_yf4L · 2023-11-23
> > > **Response to Authors**
> > >
> > > Thanks a lot for your answer, that makes sense. I keep my positive impression on the paper and will preserve my score.

---

### Official Review · Reviewer_71LW · 2023-11-12

**Soundness:** 3 good
**Presentation:** 3 good
**Contribution:** 1 poor
**Rating:** 3
**Confidence:** 4

**Summary:**

The authors present a scheme for efficient and reliable uplink communication over a noisy channel in a federated environment. In particular, clients submit rank factors of the gradients, where owing to the inherent low-rank structure of gradients, reconstruction can be performed reliably server-side, leveraging error-feedback if necessary. The authors demonstrate the effectiveness of this communication strategy, dubbed LASER, by performing experiments such as an image-classification, a GPT language-modeling task, and a 1-layer NN MNIST task.

**Strengths:**

1. The narrative and motivation are very well-written. Indeed, uplink communication is often assumed to be noiseless in many FL setups.
2. To the best of my knowledge, the theory is sound, and all assumptions are fair, with well-cited support regarding the low-rank nature of gradients.
3. The large-scale GPT-2 experiment was quite impressive. Challenging large-scale language tasks are seldom seen in FL literature.

**Weaknesses:**

1. In my opinion, this work does not introduce anything truly novel or interesting to the subdomain of communication efficiency. Gradient compression via sketching (Rabbani et al., 2023, Rothchild et al., 2020; Haddadpour et al., 2020), quantization (Zakerinia et al., 2023), and even the blanket consideration of general contraction/compression operators (Dorfman et al., 2023) along with many other works indicate that the research area of uplink efficiency is saturated. Introducing a low-rank decomposition of the gradient as a form of compression is not a particularly novel technique, especially since it is handled under the general error-feedback recovery. Bi-directional compression is a much more challenging and relevant problem in the modern landscape of FL communication efficiency.

2. The following lines concern me: "Rank compression methods (Yu et al., 2018; Cho et al., 2019; Wang et al., 2018) spectrally decompose gradient matrix (often via SVD) and transmit these factors. Since SVD is computationally prohibitive, we rely on the state-of-the-art
light-weight compressor PowerSGD (Vogels et al., 2019)." Since gradient rank decomposition has been performed in distributed settings, using a SoTA method for decomposition is simply a plug-and-play extension which further restricts the novelty of LASER.

**Questions:**

1. In what scenario would the uplink communication be noisy but the downlink communication would be noiseless? "For the downlink
communication from the server to the clients (broadcast channel), we assume that it is noiseless and thus the clients exactly receive what the server transmits..."

2.  (Chang & Tandon, 2020; Guo et al., 2020) imply that clients must first transmit information regarding the norms of their gradients before constructing a power budget policy -- does this imply 2 rounds of communication would be necessary assuming a dynamic power schedule is employed?

---

> ### Author Response · Authors · 2023-11-17
> **Novelty and communication links**
>
> We thank the reviewer for the constructive feedback and insightful comments. We refer to the common response about the novelty of LASER.
>
> - **Uplink and downlink noise:** In the uplink channel the resource-constrained workers communicate with the server whereas in the downlink communication the server to the clients. The downlink is a broadcast channel which is typically assumed in the literature to be noiseless [1-3]. The underlying intuition here is that the central server (e.g. base station) is not as resource/power-constrained as the individual clients (e.g. mobile phones) and hence it can ensure its transmission error-free relatively speaking. While few works [8, 9] study the impact of downlink noise on the convergence of Z-SGD, it’s an interesting future direction to extend these results for LASER-like compression algorithms.
>
> - **Transmission of power scalars:** Yes, indeed. As highlighted in [4, 5], the nodes transmit only the gradient norms $\|\| \boldsymbol{g}_i \|\|$, which are scalars, to the server which then computes their maximum and transmits it back. Since this procedure only involves transmission of real scalars, it has negligible communication overhead as compared to the gradients being transmitted.
>
> **References:**
>
> -  [1] H Brendan McMahan and Daniel Ramage. Federated learning: Collaborative machine learning without centralized training data. *https://ai.googleblog.com/2017/04/federated-learning-collaborative.html*, 2017.
>
>   - [2] Jakub Koneˇcn `y, H Brendan McMahan, Felix X Yu, Peter Richtárik, Ananda Theertha Suresh, and Dave Bacon. Federated learning: Strategies for improving communication efficiency. *arXiv preprint arXiv:1610.05492*, 2016.
>
>    - [3] Jakub Koneˇcn `y, H Brendan McMahan, Daniel Ramage, and Peter Richtárik. Federated optimization: Distributed machine learning for on-device intelligence. *arXiv preprint arXiv:1610.02527*, 2016.
>
> - [4] Huayan Guo, An Liu, and Vincent KN Lau. Analog gradient aggregation for federated learning over wireless networks: Customized design and convergence analysis. *IEEE Internet of Things Journal*, 8(1):197–210, 2020.
>
> - [5] Wei-Ting Chang and Ravi Tandon. Mac aware quantization for distributed gradient descent. *In GLOBECOM 2020-2020 IEEE Global Communications Conference*, pages 1–6. IEEE, 2020.

---

### Author Response · Authors · 2023-11-17
**Novelty of LASER and efficiency of rank-decomposition**

We thank all the reviewers for the encouraging feedback and valuable suggestions. We address the common concerns below.

## Novelty of LASER (71LW, ykHv, DPBP, dVrm)

We would like to emphasize that LASER is the first gradient compression algorithm (to the best of our knowledge) with theoretical guarantees to achieve state-of-the-art results with resource-constrained noisy channels, especially for the challenging GPT language modeling. While LASER takes advantage of known techniques like rank compression, it is not a simple plug-and-play combination of them. In particular, it is challenging to combine the right ingredients to design a practical compressor especially for complex tasks in computer vision and language modeling. For example, the well-known A-DSGD [1] relies on sparsification and approximate-message-passing algorithms, but does not scale beyond MNIST-1-layer-NN to tasks of practical interest.

More technically, (i) though gradient compression for distributed optimization is well-studied, most of the works here assume no communication noise and in presence of noise they perform significantly worse, as demonstrated in Section 4 of the paper; (ii) while the power-constrained channel model behind LASER is same as that of [2], the latter doesn’t employ any compression and the results restricted a to 1-layer NN and are hard to scale to complex practical tasks.

On the theoretical front, since LASER relies on the error-feedback its convergence guarantees resemble that of the classical error-feedback algorithms with no communication noise. However, in the presence of noise and power-constraints, the analysis is not straightforward and is challenging because (i) the channel noise is not a simple additive term, but a non-linear operation owing to the power constraints (Eq. (noisy channel)). To this end, our main contribution is to capture the interplay of the channel and compression through a novel quantity **channel-influence factor**. In addition to allowing a principled convergence analysis, the channel-influence factor also provides a lens to compare various compression schemes (e.g. $\lambda_{\text{LASER}}$ and $\lambda_{\text{Z-SGD}}$) and generalizes the results of the noiseless setting; (ii) to obtain a tight characterization of the channel influence factor, appropriate power allocation has to be done across various rank-factors. Here our main theoretical contribution is to show that the division of energy proportional to square root of the singular values yields an optimal power allocation scheme (Lemmas 7 and 8 in Appendix B.1), which could be of independent theoretical interest.

In view of this, we believe the contributions of LASER are significant.

## Efficiency of rank decomposition (t6dq, DPBP)

As already demonstrated in Section 4.3 of the paper, rank-decomposition of LASER is indeed extremely efficient and practical. Since LASER relies on PowerSGD [3] for the rank compression, it inherits the same low-complexity benefits: Tables 3-7 of [3] demonstrate that PowerSGD is efficient with significantly lower computational needs and has much smaller processing time/batch as compared to baselines (Top-K, Random-K) without any accuracy drop. In fact, it is the core distributed algorithm behind the recent breakthrough DALL-E (Appendix E in [4]).

**References:**

- [1] Mohammad Mohammadi Amiri and Deniz Gündüz. Machine learning at the wireless edge: Distributed stochastic gradient descent over-the-air. IEEE Transactions on Signal Processing, 68:2155–2169, 2020.

- [2] Huayan Guo, An Liu, and Vincent KN Lau. Analog gradient aggregation for federated learning over wireless networks: Customized design and convergence analysis. *IEEE Internet of Things Journal*, 8(1):197–210, 2020.

- [3] Thijs Vogels, Sai Praneeth Karimireddy, and Martin Jaggi. PowerSGD: Practical low-rank gradient compression for distributed optimization. *Advances in Neural Information Processing Systems*, 32, 2019.

- [4] Ramesh, Aditya, Pavlov, Mikhail, Goh, Gabriel, Gray, Scott, Voss, Chelsea, Radford, Alec, Chen, Mark, Sutskever, Ilya, 2021. Zero-shot text-to-image generation. *arXiv:2102.12092, [cs.CV]*.

---

> ### Comment · Reviewer_ykHv · 2023-11-18
>
> Thanks for the response. Regarding novelty in the analysis, it would be helpful if you could point to specific steps (e.g., equation numbers) where the analysis is different from standard error-feedback analysis.
>
> My following comments remain unaddressed:
> - The considered wireless communication system with "over the air" averaging is essentially doing the averaging in the analog domain. This is impractical as all modern wireless systems are digital and data transmission involves channel coding, which is likely incompatible with "over the air" averaging. I am aware that this concept has been presented in many papers published in wireless communications journals and conferences, but it is still impractical. It is unlikely that wireless communication standards will incorporate a specific physical layer technique only for the sake of computing averages, which in any foreseeable future represents only a very small fraction of data transmitted in common wireless systems. Moreover, this mechanism restricts the averaging to clients communicating with a single basestation, while in practice, the coverage of a single basestation is quite small and having all the clients connecting to the same basestation is very unlikely.
> - It is not quite clear how power reduction is achieved as shown in the experiments. The power budget is fixed according to Algorithm 1. If there is a fixed power budget, the same budget should apply to both the proposed algorithm and the baselines. The results in Table 1 and Table 3 either do not enforce a fixed power budget, which contradicts with the description in Algorithm 1, or there is something else wrong.

---

> ### Author Response · Authors · 2023-11-18
> **Feedback analysis and power budget**
>
> We apologize for the delay in our response. Regarding the compatibility with the wireless communication standards and base stations, we responded to your review directly. Here we address the remaining questions:
>
> - **Feedback analysis:** Here the main idea is to control the influence of the channel via the channel influence factor to leverage the standard error-feedback analysis. To this end, in Appendix B, as highlighted in Eq. (LASER), Appendix B.1 and B.2 obtain characterizations of the channel influence factor. Building upon this, in B.3, we utilize the standard error-feedback techniques taking this new factor into account and incorporating it into the standard lemmas for error bounds and the progress in the descent step (Lemmas 11-13).
>
> - **Power budget:** Please note that the power budget is fixed across all the baselines for comparison. Then this fixed budget is varied in order to capture the relationship between the final accuracy and the power budget used (accuracy improves with more power). Hence the tables 1 and 3 show that for the baseline Z-SGD to achieve the same accuracy as that of LASER, it needs $16 \times$ more power.

---

> > ### Comment · Reviewer_ykHv · 2023-11-21
> >
> > Thanks for the answers. I'm happy to raise my score to 6.

---

> > > ### Author Response · Authors · 2023-11-23
> > > **Thank you!**
> > >
> > > We thank the reviewer for the constructive and valuable feedback and reconsideration of the score!

---

### Author Response · Authors · 2023-11-20
**Author-reviewer discussion period ending soon**

Dear reviewers,

Thank you for your constructive feedback and comments. Since the discussion period ends on Nov 23rd, we request you to please read our rebuttal and update the score if your concerns have been addressed. We are happy to engage in further discussion for any follow-up comments.

---

### Meta-Review · Area_Chair_v5cy · 2023-12-13

**Metareview:**

The paper presents a novel low-rank compression approach tailored for distributed optimization, with a particular emphasis on its application in wireless communication systems. In this context, the averaging process takes place "over the air" within a noisy channel. The proposed method is evaluated on a variety of tasks, including language modeling and image classification.

Strengths: The paper is well-written in terms of its narrative and motivation. The authors provide a comprehensive explanation of their method, and in particular, the beginning of Section 3 is well-written and makes it easy for readers to understand the main techniques. The large-scale GPT-2 experiment is impressive, and challenging large-scale language tasks are rarely seen in FL literature.

Weaknesses: Reviewer 71LW's first question, "In what scenario would the uplink communication be noisy but the downlink communication would be noiseless?", seems to be a valid question. The authors' response does not provide a strong motivation for this assumption or provide a good example of such a scenario.Without a solid rationale for this assumption, the superior performance of the proposed algorithm in the noisy setting (as shown in the experimental section) is not significant.

Reviewer 71LW stated in Weakness 1, "In my opinion, this work does not introduce anything truly novel or interesting to the subdomain of communication efficiency. Gradient compression via sketching (Rabbani et al., 2023, Rothchild et al., 2020; Haddadpour et al., 2020), quantization (Zakerinia et al., 2023), and even the blanket consideration of general contraction/compression operators (Dorfman et al., 2023) along with many other works indicate that the research area of uplink efficiency is saturated." The methods mentioned by reviewer 71LW are indeed widely used baselines. As such, the authors' experiments are not as strong as they could be, since they do not compare to these methods.

**Justification For Why Not Higher Score:**

See the weaknesses above.

**Justification For Why Not Lower Score:**

N/A

---

### Decision · Program_Chairs · 2024-01-16

Reject